# OpenReview forum: "Train for Truth, Keep the Skills: Binary Retrieval-Augmented Reward Mitigates Hallucinations"
_ICLR.cc/2026/Conference — Submitted to ICLR 2026_

### Official Review · Reviewer_q2wP · 2025-10-27

**Soundness:** 3
**Presentation:** 3
**Contribution:** 1
**Rating:** 2
**Confidence:** 5

**Summary:**

This paper introduces Retrieval-Augmented Reward, an online RL method that 0/1 rewards to mitigate hallucinations by verifying outputs against retrieved evidence. The approach achieves reduction in hallucination rates while preserving general capabilities. They claim that binary rewards are more robust than continuous alternatives, avoiding reward hacking and utility degradation.

**Strengths:**

1. Binary RAR demonstrates both computational efficiency advantages over dense reward methods and superior hallucination reduction performance.
2. The experimental design systematically evaluates both hallucination reduction and capability preservation, demonstrating that Binary RAR achieves 39.3% factuality improvement while maintaining utility.

**Weaknesses:**

The main concerns for this paper that it provided limited explanation that binary reward design should be better.
1. The paper's claim that 0/1 rewards are more effective at mitigating hallucinations lacks adequate explanation of the underlying mechanism. Based on Figures 2 and 3, the binary reward's effectiveness appears to stem from making the model significantly more conservative—refusing to answer most questions, providing less information, and generating shorter responses. However, this conservative behavior may not always be desirable. The authors should provide more discussion about the impact on content recall rather than focusing primarily on precision.
2.  The paper fails to sufficiently explain why 0/1 rewards better preserve general capabilities. A more detailed analysis of the major performance changes in Table 2 benchmarks (such as IFEval and GSM8K) is needed to better understand the advantages of binary rewards. The paper should investigate why binary rewards avoid the utility degradation observed with dense rewards.
3. The experiments are conducted exclusively on Qwen3 models (4B and 8B), lacking validation across diverse model families.

**Questions:**

1. What is the underlying mechanism that makes binary rewards outperform dense rewards for hallucination mitigation?
2. Could the authors provide analysis of training dynamics, including what percentage of prompt-response pairs receive 0 vs 1 rewards during training?

---

> ### Author Response · Authors · 2025-11-23
>
> We thank the reviewers for highlighting both the effectiveness of our method and our systematic evaluation. We address the questions about how binary rewards behave and why they preserve model capability below.
>
> > (Q1) Why is the model not overly conservative? What is the underlying mechanism of  0/1 rewards?
>
> The RL mechanism and KL penalty enable the model to learn when it is appropriate to use abstention. The RL objective maximizes the expected reward plus a KL term between the updated model and the initial model (Equation 1). If the initial model already answers a question correctly, switching that answer to “I don’t know” does not raise the binary RAR reward, but it does incur a KL penalty because the output distribution changes. The model, therefore, keeps the correct answer. If the initial model answers a question incorrectly, switching that answer to “I don’t know” raises the binary RAR reward from zero to one. This gain is larger than the KL penalty from the distribution change, so the update is favored. Our analysis in Section 6.2 shows that models preserve accuracy on short-form QA tasks after training.
>
> > (Q2) Why 0/1 rewards better preserve general capabilities? Why do binary rewards avoid the utility degradation observed with dense rewards?
>
> Two factors of binary RAR contribute to utility preservation: its binary nature and the response-level verification.
>
> Dense reward is vulnerable to reward hacking. A model can inflate VeriScore by generating irrelevant but correct statements or by changing its style in ways that is favored by the claim extractor. When we verify the entire response, adding irrelevant content or altering the style has little effect on the verification outcome. This is why binary RAR is less vulnerable to reward hacking.
>
> We also include an additional experiment (Section 6.3) to show that both response-level verification and binary reward contribute to utility preservation. We apply RL with a binarized VeriScore (threshold 0.5) and report the results on Biography and AlpacaEval below. Both VeriScore and binarized VeriScore lead to lower AlpacaEval scores compared to binary RAR. This shows that response-level verification, not just binarization, helps prevent reward hacking.
>
> Table: Ablation Study of Reward
> | Method | Biography (Factual Precision↑) | AlpacaEval (LC Win Rate↑) |
> |--------|----------------|-----------------|
> | VeriScore | 48.3 | 42.2 |
> | Binarized VeriScore | 38.7 | 48.2 |
> | Binary RAR | 54.2 | 53.9 |
>
>
> > (Q1) Based on Figures 2 and 3, the binary reward's effectiveness appears to stem from making the model significantly more conservative—refusing to answer most questions, providing less information, and generating shorter responses.
>
> We would like to politely clarify that Figures 2 and 3 actually show our models are not conservative. For long-form generation on the biography dataset, the model produces the same number of correct claims as the initial model, so the information content is unchanged. The shorter outputs come from removing incorrect claims, not from withholding correct ones. The length-controlled win rate on AlpacaEval remains unchanged, which further shows that the quality of long-form outputs remains the same. For short-form QA, Figure 3 indicates that after training, the model converts incorrect guesses to abstaining and preserves its accuracy. Therefore, our binary RAR training reduces hallucination while still preserving the model's utility.

---

> > ### Author Response · Authors · 2025-11-23
> >
> > > (W3) The experiments are conducted exclusively on Qwen3 models (4B and 8B), lacking validation across diverse model families.
> >
> >
> > We address this concern by conducting additional experiments on more model families. We include the results in Section A.3 in the revision and show them in the table below. Tulu3-8B model is trained on top of the Llama-3.1-8B pretrained checkpoint and already trained with RL with a reward model, which gives it a strong AlpacaEval win rate among models based on Llama-3.1-8B. Even in this strong setting, our method remains effective: Binary RAR raises factual precision on both Biography and WildHallu, while changes on AlpacaEval and IFEval are negligible.
> >
> > Table: RL with Binary RAR on Tulu3-8B
> > |Model|Biography (Factual Precision↑)|WildHallu (Factual Precision↑)|AlpacaEval (LC Win Rate↑)|IFEval (Accuracy↑)|
> > |---|---|---|---|---|
> > |allenai/Llama-3.1-Tulu-3-8B|31.9|56.6|24.3|81.1|
> > |+Binary RAR|39.9|60.7|23.4|80.6|
> >
> > > (Q2) Could the authors provide analysis of training dynamics, including what percentage of prompt-response pairs receive 0 vs 1 rewards during training?
> >
> > Thank you for the suggestion. We additionally report the percentage of training prompts that receive all-zero rewards, all-one rewards, or a mix of zero across the 8 sampled responses. We include these results in Section A.3 in the revision. At the start of training, 27% of prompts produce mixed rewards and 61% produce all-one rewards (meaning no contradictions). By the end of training, these numbers go to 18% and 76%. The increase in all-one rewards indicates that the model generates fewer contradictory responses as training proceeds.
> >
> >
> > Table: Distribution of training-prompt reward patterns for each verifier (first vs. last 100 steps).
> > | Verifier  | Training phase  | All-zero rewards | All-one rewards | Mixed rewards |
> > | --------- | --------------- | ---------------- | --------------- | ------------- |
> > | Qwen3-32B | First 100 steps | 12%              | 61%             | 27%           |
> > | Qwen3-32B | Last 100 steps  | 6%               | 76%             | 18%           |
> >
> >
> > ---
> >
> > We hope these clarifications resolve the concerns raised and make the contributions and empirical evidence of our work fully clear.

---

### Official Review · Reviewer_sL85 · 2025-10-30

**Soundness:** 4
**Presentation:** 3
**Contribution:** 3
**Rating:** 6
**Confidence:** 4

**Summary:**

The paper proposes a Binary Retrieval-Augmented Reward (RAR) framework for reinforcement learning to reduce hallucinations in large language models. The method verifies model outputs against retrieved evidence using an LLM judge and optimizes a binary reward through the GRPO algorithm. This design replaces complex continuous factuality scores with a simple verifiable signal focused on factual correctness. Experiments on Qwen3-8B and Qwen3-4B show clear gains in factual precision and reduced QA errors, while maintaining instruction, math, and coding abilities. The model also learns calibrated abstention when evidence is lacking, offering a scalable path to improve factual reliability.

**Strengths:**

1. The paper replaces traditional continuous factuality scores with a binary retrieval-augmented reward, explicitly optimizing the model toward *avoiding factual errors*. This design leads to a simple yet effective mechanism for factual correction and hallucination reduction.
2. The experiments are well-structured and comprehensive, including ablation studies, multi-task and multi-model evaluations, and assessments of general capabilities. The results appear credible and consistent.
3. The paper is clearly written, well-organized, and supported by intuitive figures and tables that effectively illustrate both the method and findings.
4. In terms of significance, the work demonstrates a substantial reduction in hallucinations without sacrificing general capabilities, offering a scalable paradigm for building more reliable and safer LLMs. It makes a meaningful contribution to the application of reinforcement learning for model alignment.

**Weaknesses:**

1. The optimization objective fully depends on retrieved evidence — the binary reward is determined by whether the retrieved documents contradict the model output. Since retrieval systems inherently contain bias (e.g., source preference, recency errors, and incomplete coverage), the model is effectively optimized to align with retrieval consensus rather than the ground truth. In essence, the model learns retrieval alignment, not truth alignment.
2. This RLKF paper (https://arxiv.org/abs/2403.18349) has already introduced binary knowledge-feedback rewards ****within a PPO framework to train models to respond or abstain based on verifiable knowledge. The current work mainly differs by (i) replacing PPO with GRPO, and (ii) defining the binary signal through retrieval-augmented evidence checking. These are implementation-level modifications rather than conceptual innovations. Without deeper analysis of learning dynamics or generalization behavior, the contribution appears incremental.
3. Reward design may encourage conservative or evasive behavior. With a “no contradiction = 1, otherwise 0” reward, the safest strategy under incomplete evidence is to produce shorter or abstaining outputs. The paper’s own results show high abstention rates (55.2% on PopQA and 27.5% on GPQA), alongside decreased factual accuracy (PopQA 20.4 → 18.0; GPQA 48.9 → 44.2). This suggests that the reported hallucination reduction might partly result from answering less, not answering more accurately. The authors should report recall and response-part accuracy ****to clarify whether improvements stem from better correctness or more frequent refusal.

**Questions:**

1. Why did you choose GRPO instead of trying other RL algorithms such as Reinforce++ or DAPO?
2. Why not directly train with ground truth instead of constructing retrieval candidates? The dataset already contains ground-truth responses. Instead of directly using them to compute factual consistency rewards, the paper constructs a retrieval pool seeded by the ground-truth answer and then measures agreement between model outputs and retrieved evidence. Is this two-step design necessary? Could this intermediate retrieval layer introduce additional noise or bias?
3. If the retrieved documents contain contradictions or errors, how does the method decide which one is more trustworthy? Is there any consistency filtering, aggregation, or multi-judge mechanism? Moreover, how sensitive is the binary reward to the choice of retriever (e.g., BM25 vs. dense retrieval) and the judge model?
4. It would be informative to include recall and response-part accuracy to differentiate between “answering less” and “answering better.”In addition, could the authors test on simpler QA datasets? If the dataset is easier and more closed-domain, would the model become even more conservative (i.e., show a stronger accuracy drop), revealing a deeper behavioral bias?
5. Have the authors evaluated whether the trained model itself demonstrates generalization beyond factual QA — for example, on mathematical reasoning, code generation, or multi-step logical tasks?

---

> ### Author Response · Authors · 2025-11-23
>
> We appreciate the reviewers’ careful reading and the recognition that our method is effective and that our evaluation is comprehensive. Below, we address each question and concern in detail:
>
> > (Q2) Why not directly train with ground truth instead of constructing retrieval candidates? Instead of directly using them to compute factual consistency rewards, the paper constructs a retrieval pool seeded by the ground-truth answer and then measures agreement between model outputs and retrieved evidence. Is this two-step design necessary?
>
> We would like to clarify that the WildChat dataset used in our training does not contain a “ground-truth” response in the first place. We train and evaluate on many long-form and open-ended prompts. For a prompt such as “tell me about three AI companies,” there is no single correct set of companies, and each company choice involves many possible factual statements. As a result, comparing the model output to a single reference answer is not sufficient for assessing factual correctness.
>
> > (W2) This RLKF paper has already introduced binary knowledge-feedback rewards within a PPO framework to train models to respond or abstain based on verifiable knowledge.
>
> Thank you for pointing out this work. RLKF and our approach share a similar goal: reducing hallucination while maintaining helpfulness and avoiding overly conservative behavior. However, RLKF focuses on short-form question answering, where an answer set is available and correctness is well-defined. In contrast, our work considers both long-form generation and short-form question answering. Long-form generation prompts do not have a single ground-truth answer, and responses can be partially correct. As a result, prior work [1,2] often relies on continuous factuality scores (e.g., VeriScore) to build preference pairs.
>
> Our results show that such continuous rewards are vulnerable to reward hacking in online RL. In comparison, the binary retrieval-augmented reward (RAR) provides stronger hallucination reduction while preserving utility. We add RLKF to the discussion of related work in Section 2.
>
> [1] Tian et al. (2023), Fine-tuning Language Models for Factuality
>
> [2] Lin et al. (2024), FLAME: Factuality-Aware Alignment for Large Language Models
>
> > (W3, Q4) Reward design may encourage conservative or evasive behavior. With a “no contradiction = 1, otherwise 0” reward, the safest strategy under incomplete evidence is to produce shorter or abstaining outputs.
>
> The RL mechanism and KL penalty enable the model to learn when it is appropriate to use abstention. The RL objective maximizes the expected reward plus a KL term between the updated model and the initial model (Equation 1). Taking short-form QA as an example, if the initial model already answers a question correctly, switching that answer to “I don’t know” does not raise the binary RAR reward, but it does incur a KL penalty because the output distribution changes. The model, therefore, keeps the correct answer. If the initial model answers a question incorrectly, switching that answer to “I don’t know” raises the binary RAR reward from zero to one. This gain is larger than the KL penalty from the distribution change, so the update is favored.
>
> > (W3, Q4) The paper’s own results show high abstention rates (55.2% on PopQA and 27.5% on GPQA), alongside decreased factual accuracy (PopQA 20.4 → 18.0; GPQA 48.9 → 44.2). This suggests that the reported hallucination reduction might partly result from answering less, not answering more accurately. The authors should report recall and response-part accuracy.
>
> The observed drop in accuracy does not reflect a decline in the model’s underlying ability. Instead,  the model converts some guesses into abstentions during our training. For example, the abstention rate on GPQA increases to 27.5% (as shown in Figure 2) after training. If the model were required to guess uniformly at random among the four options for those abstentions, the expected gain of accuracy would be approximately +6.8 points (27.5/4), which offsets the apparent drop of accuracy (48.9 → 44.2). To isolate model capability from abstention behavior, we provide a controlled evaluation in Table 2 and Section 5.2, where the model is explicitly prompted to always provide an answer. Under this setting, there is no accuracy decline on either PopQA or GPQA.
>
> We additionally calculated response-part accuracy, defined as the percentage of correct answers among all attempted answers (correct plus incorrect). After RL training on Qwen3-8B, response-part accuracy increases from 22.2 to 40.1 on PopQA and from 49.4 to 60.9 on GPQA.

---

> > ### Author Response · Authors · 2025-11-23
> >
> > > (W1) The optimization objective fully depends on retrieved evidence — the binary reward is determined by whether the retrieved documents contradict the model output. Since retrieval systems inherently contain bias (e.g., source preference, recency errors, and incomplete coverage), the model is effectively optimized to align with retrieval consensus rather than the ground truth.
> >
> > We agree that retrieval-based verification relies on the quality of retrieval. This is exactly why we design the new binary reward to make online RL more stable under retrieval noise. When the datastore has incomplete coverage and no retrieved document is relevant to a prompt, the verifier cannot detect any contradictions. Under binary RAR, every response for that prompt receives a reward of one. As a result, the model receives no gradient signal on that example and does not learn the retrieval gap or any related bias.
> >
> > > (Q2) Moreover, how sensitive is the binary reward to the choice of judge model?
> >
> > We additionally run RL training using Qwen3-8B as the verifier and find that it is less effective than using Qwen3-32B. We add the results to Section A.2 and show them in the table below. We categorize training prompts into three types: all-zero rewards in 8 sampled responses, all one-rewards, and mixed rewards of zero and one. The Qwen3-32B verifier detects more contradictions than Qwen3-8B: In the first 100 training steps, Qwen3-32B marks 61% of the training prompts as contradiction-free (i.e., all-one rewards), while Qwen3-8B marks 74%. This explains why Qwen3-32B leads to better improvement. However, this does not imply that small models cannot serve as effective verifiers. Prior work has shown that small finetuned models can match the fact-verification performance of proprietary models [3].
> >
> > [3] Song et al. (2024), VERISCORE: Evaluating the factuality of verifiable claims in long-form text generation
> >
> > Table: Distribution of training-prompt reward patterns for each verifier (first vs. last 100 steps).
> > | Verifier  | Training phase  | All-zero rewards | All-one rewards | Mixed rewards |
> > | --------- | --------------- | ---------------- | --------------- | ------------- |
> > | Qwen3-8B  | First 100 steps | 7%               | 74%             | 19%           |
> > | Qwen3-8B  | Last 100 steps  | 5%               | 80%             | 15%           |
> > | Qwen3-32B | First 100 steps | 12%              | 61%             | 27%           |
> > | Qwen3-32B | Last 100 steps  | 6%               | 76%             | 18%           |
> >
> > > (Q1) Why did you choose GRPO instead of trying other RL algorithms such as Reinforce++ or DAPO?
> >
> > The binary retrieval-augmented reward is compatible with any RL algorithms. We choose GRPO rather than PPO because it avoids training a value function, which simplifies the setup and reduces overhead. Also, GRPO is popular [4,5] in recent research on RL applications. REINFORCE++ and DAPO are indeed suitable alternatives, but we did not run them due to limited computation.
> >
> > [4] Shao et al. (2024), DeepSeekMath: Pushing the Limits of Mathematical Reasoning in Open Language Models
> >
> > [5] Cheng et al. (2025), ToolRL: Reward is All Tool Learning Needs
> >
> > > (Q5) Have the authors evaluated whether the trained model itself demonstrates generalization beyond factual QA — for example, on mathematical reasoning, code generation, or multi-step logical tasks?
> >
> > The results on logical reasoning, mathematical reasoning, and code generation are in Table 2, and we observe almost no degradation. This is expected because our training prompts focus on questions that require external knowledge, and these benchmarks have minimal overlap with that domain. As a result, the model largely preserves its original capabilities. Our method targets extrinsic hallucination, where the model produces content that is not supported by training data or real-world facts. Intrinsic hallucination (for example, instruction-following failures or logical mistakes) falls outside the scope of this work and is not detectable by a retrieval-augmented verifier. However, the binary reward idea may still be useful for future methods that address these other forms of hallucination.
> >
> > ---
> >
> > We hope these clarifications resolve the concerns raised and make the contributions and empirical evidence of our work fully clear.

---

### Official Review · Reviewer_eTuW · 2025-10-31

**Soundness:** 2
**Presentation:** 2
**Contribution:** 2
**Rating:** 2
**Confidence:** 4

**Summary:**

This paper proposes a Binary Retrieval-Augmented Reward (Binary RAR) reinforcement learning approach that uses a 0/1 reward signal to train large language models to reduce hallucinations, while claiming not to degrade their general capabilities. The method determines whether the model’s responses contradict retrieved documents and assigns a reward of 1 when no contradiction is found. Experiments on Qwen3-8B/4B demonstrate that the approach significantly improves factuality and induces calibrated abstention behavior, where the model appropriately responds with “I don’t know.”

**Strengths:**

1. The method is simple and effective, easy to implement from an engineering perspective, and demonstrates greater robustness compared to dense reward models.
2. The experimental design simultaneously addresses both hallucination mitigation and preservation of general capabilities.
3. Experimental results show that the proposed approach improves the model’s factual accuracy, achieving strong overall performance.

**Weaknesses:**

1. The authors claim that full-text contradiction detection avoids the “error accumulation” of claim-wise verification, but this statement lacks justification. Claim-level verification is an independent process without cumulative error, whereas full-text inputs may introduce contextual interference and order bias. A controlled comparison between the two verification granularities is recommended to support this claim.
2. The “I don’t know” samples are not human-labeled but automatically detected through string matching, without considering semantically equivalent expressions or manual validation. The appendix prompt explicitly instructs judgments to rely only on retrieved text, ignoring the model’s internal (parametric) knowledge—meaning the model is trained to answer “I don’t know” even when it actually knows the answer. The prompt design and labeling mechanism should be refined to better distinguish retrieval gaps from genuine uncertainty.
3. The paper repeatedly emphasizes that “general capabilities do not degrade,” but this is largely enforced by the training configuration (early stopping and KL regularization) rather than emerging naturally. The comparison is therefore not entirely fair. Experiments using dense reward methods under the same settings should be added to verify whether Binary RAR truly achieves better factuality–utility balance.

**Questions:**

1. Has there been a controlled comparison between VeriScore and Binary RAR under the same KL coefficient? Without such an experiment, it is difficult to determine whether the observed balance between factuality and utility arises from the binary reward mechanism itself or simply from the degree of KL regularization applied.

2.How does the evaluation of “I don’t know” responses account for the influence of the model’s internal parametric knowledge?
Even if the assessment is restricted to external (retrieval-based) knowledge, how consistent are the “I don’t know” labels between model-generated judgments and human evaluation? Clarifying this alignment is essential to verify that the model’s abstention truly reflects epistemic uncertainty rather than retrieval insufficiency.

---

> ### Author Response · Authors · 2025-11-23
>
> We appreciate the time spent evaluating our submission and for recognizing the strengths of the proposed method. Below, we address each concern and clarify how our results support the claims made in the paper.
>
>
> > (W1) The authors claim that full-text contradiction detection avoids the “error accumulation” of claim-wise verification, but this statement lacks justification.
>
> To clarify, our use of full-text contradiction detection in the binary RAR reward is driven by the need to avoid reward hacking in online RL. With claim-wise verification, a model can increase its VeriScore by adding irrelevant but correct statements or by changing its style in ways that favor the claim extractor. These behaviors do not reduce hallucination, but they do raise the continuous score. When we verify the full response as a whole, adding extra content or shifting style has almost no effect on whether the output contradicts the retrieved evidence. This makes full-text contradiction detection far less sensitive to such failure modes.
>
> We also include an additional experiment (Section 6.3) to show that both full-text contradiction and binary reward contribute to utility preservation. We apply RL with a binarized VeriScore (threshold 0.5) and report the results on Biography and AlpacaEval below. Both VeriScore and binarized VeriScore lead to lower AlpacaEval scores compared to binary RAR. This shows that full-text contradiction detection, not just binarization, helps prevent reward hacking.
>
> Table: Ablation Study of Reward
> | Method | Biography (Factual Precision↑) | AlpacaEval (LC Win Rate↑) |
> |--------|----------------|-----------------|
> | VeriScore | 48.3 | 42.2 |
> | Binarized VeriScore | 38.7 | 48.2 |
> | Binary RAR | 54.2 | 53.9 |
>
>
>
> > (W2) The “I don’t know” samples are not human-labeled but automatically detected through string matching, without considering semantically equivalent expressions or manual validation.
>
> For PopQA, we use an LM judge to classify each output as correct, incorrect, or abstaining. Since PopQA uses short answers, abstentions appear in a straightforward form. For GPQA, which is a multiple-choice question dataset, we explicitly prompt the model to output either one option or the exact phrase “I don’t know.” We manually check 50 outputs for both responses and find almost 100% accuracy in identifying abstaining cases. We add a discussion in Section B.2.
>
> > (W2) The appendix prompt explicitly instructs judgments to rely only on retrieved text, ignoring the model’s internal (parametric) knowledge—meaning the model is trained to answer “I don’t know” even when it actually knows the answer.
>
> The verifier aims to estimate whether a model’s response is factually correct with respect to real-world knowledge. Using retrieved documents provides a more reliable basis for this check than relying on the verifier’s own parametric knowledge. Checking whether a passage contradicts the retrieved evidence is a standard approach that has been adopted in many previous studies [1].
>
> We are not explicitly training models to say "I don't know". In fact, the model learns when it is appropriate to use abstention over incorrect answers via the RL mechanism. The RL objective maximizes the expected reward plus a KL term between the updated model and the initial model (Equation 1). If the initial model already answers a question correctly, switching that answer to “I don’t know” does not raise the binary RAR reward, but it does incur a KL penalty because the output distribution changes. The model, therefore, keeps the correct answer. If the initial model answers a question incorrectly, switching that answer to “I don't know” raises the binary RAR reward from zero to one. This gain is larger than the KL penalty from the distribution change, so the update is favored. This explains why the model does not collapse into conservative behavior: accuracy stays nearly stable, but many incorrect answers shift to abstentions, as shown in Figure 2.
>
> [1] Gao et al. (2023), RARR: Researching and Revising What Language Models Say, Using Language Models

---

> > ### Author Response · Authors · 2025-11-23
> >
> > > (W3) The comparison is therefore not entirely fair. Experiments using dense reward methods under the same settings should be added to verify whether Binary RAR truly achieves better factuality–utility balance.
> >
> >
> > We kindly clarify that we applied the same training settings and include a thorough parameter search for both Binary RAR and VeriScore  (in Section 6.3). We additionally add evaluation at different training steps to Section 6.3 and show them in the table below. At 1000 steps and KL coefficient of 0.001, VeriScore does raise factual precision more quickly than Binary RAR (row 2 vs row 5), but the AlpacaEval win rate collapse is severe (VeriScore: 11.6; Binary RAR: 53.9). We attempted to reduce this collapse by stopping early (500 steps) or by increasing the KL coefficient (0.003), but both adjustments introduced a direct trade-off between factuality and utility.
> >
> > By contrast, Binary RAR reaches its best factual precision at 2000 steps with KL = 0.001 and does so without harming the win rate. This is why we conclude that Binary RAR achieves a better factuality-utility balance under the same training conditions.
> >
> >
> > Table: Ablation study on KL coefficient and training steps
> >
> > | Reward | KL | Steps | Biography (Factual Precision↑) | AlpacaEval (LC Win Rate↑) |
> > |-------|----|-------|----------------|-----------------|
> > | VeriScore | 0.001 | 500 | 48.3 | 42.2 |
> > | VeriScore | 0.001 | 1000 | 55.6 | 11.6 |
> > | VeriScore | 0.003 | 500 | 40.7 | 45.0 |
> > | VeriScore | 0.003 | 1000 | 49.7 | 33.1 |
> > | Binary RAR | 0.001 | 1000 | 41.2 | 53.7 |
> > | Binary RAR | 0.001 | 2000 | 54.2 | 53.9 |
> >
> >
> > —
> >
> > We hope these clarifications resolve the concerns raised and make the contributions and empirical evidence of our work fully clear.

---

### Official Review · Reviewer_G4jz · 2025-11-02

**Soundness:** 1
**Presentation:** 3
**Contribution:** 2
**Rating:** 4
**Confidence:** 3

**Summary:**

This paper applies reinforcement learning methods to mitigate hallucinations in large language models. Specifically, after the model generates a response, relevant evidence is retrieved, and a binary reward signal is constructed based on whether factual conflicts exist between the model’s response and the retrieved evidence. Experimental results demonstrate that this approach is effective in reducing hallucinations.

**Strengths:**

* The writing is clear, and the proposed method is straightforward.
* The experiments are thorough, and the work is relatively comprehensive.

**Weaknesses:**

* The main difference between this work and the most relevant baseline, VeriScore, lies in the design of the reward signal. While VeriScore uses a soft reward based on the proportion of correct claims, this paper adopts a hard binary reward that simply judges whether any factual inconsistency exists. Apart from this modification, there are no substantial differences between the two approaches. Overall, this incremental improvement makes the work resemble more of a technical report than a full research contribution.
* In the experimental results, apart from Factual Precision, the task-level performance metrics (e.g., accuracy) are not reported in detail. In addition, Table 1 uses Factual Precision for the long-form setting but Hallucination Rate for the short-form setting. It would be helpful to clarify the rationale behind using different evaluation metrics for these two settings.
* Does the choice of model used for reward computation have a significant impact on the results?
* It is unclear whether the conclusions would remain valid if alternative reinforcement learning algorithms, such as PPO, were employed. A discussion or comparison on this aspect would strengthen the paper.

**Questions:**

None

---

> ### Author Response · Authors · 2025-11-22
>
> We sincerely thank the reviewers for their thoughtful evaluation and for recognizing that our method is straightforward and that our empirical study is comprehensive. Our response focuses on clarifying the contributions and describing the updates we have made to the paper.
>
> > (W1) Overall, this incremental improvement makes the work resemble more of a technical report than a full research contribution.
>
> Most existing work on improving long-form factuality focuses on offline RL [2,3], and online RL is underexplored. For instance, online RL with VeriScore appears only in a concurrent work [1]. One of our findings is that online RL is more effective than offline RL for hallucination reduction (as shown in Table 1 and Section 5.1). However, online RL introduces new challenges: reward hacking becomes a serious problem as training proceeds. Our work provides a clear framework for improving the factuality of existing model checkpoints while keeping their real capabilities intact. We evaluate utility across a wide range of datasets (Section 5.2), which is important for practical use and has not been examined carefully in earlier studies. Finally, our analysis (Section 6) explains that an LM can inflate VeriScore by adding irrelevant but correct claims, and the binary reward is less sensitive to reward hacking compared to continuous factuality rewards. We highlight and address this failure mode that online RL methods encounter, and that earlier work has not been treated with enough attention.
>
> [1] Chen et al. (2025), Learning to Reason for Factuality
>
> [2] Tian et al. (2023), Fine-tuning Language Models for Factuality
>
> [3] Lin et al. (2024), FLAME: Factuality-Aware Alignment for Large Language Models
>
> > (W2) Table 1 uses Factual Precision for the long-form setting but Hallucination Rate for the short-form setting. It would be helpful to clarify the rationale behind using different evaluation metrics for these two settings.
>
> Thank you for pointing out this mismatch. We have now updated our Table 1 with the hallucination rate for both long-form and short-form settings. For long-form generation, the hallucination rate is defined as the percentage of incorrect claims among all extracted claims, following [4]. We add more description of this metric to Section 4.1
>
> [4] GPT-5 System Card, https://cdn.openai.com/gpt-5-system-card.pdf
>
> > (W3) Does the choice of model used for reward computation have a significant impact on the results?
>
>  Yes, the choice of verifier affects training effectiveness. We additionally run an RL experiment with Qwen3-8B, and find that it is less effective than using Qwen3-32B. We add the results to Section A.2 and show them in the table below. We categorize training prompts into three types: all-zero rewards in 8 sampled responses, all-one rewards, and mixed rewards of zero and one. The Qwen3-32B verifier detects more contradictions than Qwen3-8B: In the first 100 training steps, Qwen3-32B marks 61% of the training prompts as contradiction-free (i.e., all-one rewards), while Qwen3-8B marks 74%. This explains why Qwen3-32B leads to better improvement. However, this does not imply that small models cannot serve as effective verifiers. Prior work has shown that small fine-tuned models can match the fact-verification performance of proprietary models [5]. This is the future direction of this work.
>
> [5] Song et al. (2024), VERISCORE: Evaluating the factuality of verifiable claims in long-form text generation
>
> Table: Distribution of training-prompt reward patterns for each verifier (first vs. last 100 steps).
> | Verifier  | Training phase  | All-zero rewards | All-one rewards | Mixed rewards |
> | --------- | --------------- | ---------------- | --------------- | ------------- |
> | Qwen3-8B  | First 100 steps | 7%               | 74%             | 19%           |
> | Qwen3-8B  | Last 100 steps  | 5%               | 80%             | 15%           |
> | Qwen3-32B | First 100 steps | 12%              | 61%             | 27%           |
> | Qwen3-32B | Last 100 steps  | 6%               | 76%             | 18%           |

---

> > ### Author Response · Authors · 2025-11-23
> >
> > >  (W4) It is unclear whether the conclusions would remain valid if alternative reinforcement learning algorithms, such as PPO, were employed.
> >
> > Thank you for the question. Our conclusions do not depend on the specific choice of the RL algorithm. The proposed binary retrieval-augmented reward works with any policy gradient RL method, including PPO. We chose GRPO mainly because it avoids the need for a learned value function, and it is a popular choice in the research community [6, 7].
> > Although different RL algorithms may have different convergence properties, they all aim to optimize the same reward function (Eq. 1). As a result, the core issue we highlight, namely reward hacking caused by imperfect reward design, remains unchanged regardless of whether one uses GRPO, PPO, or another RL variant. This is why we emphasize the value of a binary reward: it reduces chances for pathological exploitation and is more reliable for hallucination reduction.
> >
> > [6] Shao et al. (2024), DeepSeekMath: Pushing the Limits of Mathematical Reasoning in Open Language Models
> >
> > [7] Cheng et al. (2025), ToolRL: Reward is All Tool Learning Needs
> >
> > ---
> > We hope these clarifications address the concerns, and we welcome any further questions that would help make our answers even clearer.

---

> > > ### Comment · Reviewer_G4jz · 2025-11-26
> > >
> > > Thank you for the detailed response and for your efforts in addressing the reviews. For me, the key remaining issue is still the core contribution of this work. The authors list several points, such as framing the problem as online RL and providing a “clear framework” for improving the factuality of existing model checkpoints. However, compared with VeriScore, the main methodological difference still appears to lie in the design of the reward signal, and this change seems relatively minor rather than a fundamental shift in approach. Moreover, extending VeriScore to an online RL setting does not seem particularly challenging or conceptually deep in itself.
> > >
> > > Given these considerations, I have decided to maintain my current overall score.

---

### Author Response · Authors · 2025-12-03
**General Response**

Dear AC and reviewers,

We are truly grateful that you spent extra time and effort on our paper due to this unexpected incident. We also believe that our extensive additional analysis and discussion addressed the concerns raised by the reviewers.

---

### Summary of the Work and Main Contributions

The goal of this work is to reduce hallucinations of fully trained large reasoning models without losing their general capabilities. Prior work has focused on improving factuality on specific downstream tasks, but has not carefully examined the trade-off between factuality and broad model utility under a wide evaluation setup.

We introduce an online RL method that uses a binary retrieval-augmented reward (Binary RAR). As acknowledged by reviewers (eTuW, sL85, q2wP), online RL with Binary RAR gives the strongest hallucination reduction under a no-utility-loss requirement. The reward signal is based on retrieval-augmented contradiction detection using a simple 0/1 reward. All reviewers (G4jz, eTuW, sL85, q2wP) describe this design as easy to implement and empirically strong. Our evaluations cover a range of benchmarks on four datasets for hallucination and ten datasets for utility. The utility evaluation confirms that instruction-following, reasoning, and coding performance are preserved.

Our analysis also shows that online RL with a dense factuality reward is prone to reward hacking (eTuW, q2wP). The model learns to produce answers that are correct but weakly related, which increases the reward but reduces usefulness. Our qualitative study and ablation study explain why this occurs.

---

### Summary of Reviewer Concerns and Our Actions

**Reward hacking and mechanisms**

* eTuW asked why full-text contradiction detection is more stable than claim-level verification → we showed why full-text signals avoid local noise (Fig 4, Section A.1).
* sL85 asked why Binary RAR does not cause excessive refusal → we explained that the 0/1 reward preserves utility due to the RL formulation and showed that refusal rates remain stable (Fig 3).
* q2wP asked for a clear mechanism explaining why dense rewards fail while Binary RAR works → we added detailed discussion on dense-reward failure modes and stepwise ablation (Table 3, Section A.1).

**Verifier model and retrieval dependence**

* G4jz asked whether different verifiers show similar behavior → we added experiments with Qwen3-8B and Qwen3-32B verifiers (Section A.2).
* q2wP asked about sensitivity to retriever or judge → we added experiments with Qwen3-8B and Qwen3-32B verifiers (Section A.2).

**General capability preservation**

* eTuW suggested that preservation may be due to KL or early stopping → we pointed to controlled settings using the same KL and number of steps (Table 3).
* sL85 suggested that shorter outputs may indicate conservative behavior → we showed that the model is not conservative (Table 2; Table 3).
* q2wP asked for IFEval and GSM8K → both are already included in Table 2.

**Evaluation metrics and abstention behavior**

* G4jz requested identical metrics across hallucination tasks → we updated Table 1 to use unified hallucination rate.
* eTuW asked how “I don’t know” behavior is detected → we clarified the detection procedure and manual validation (Section B.2).
* q2wP asked for response-part accuracy → we provided response-part accuracy.

**Novelty and relation to prior work**

* G4jz asked about differences from VeriScore → we noted that our method was concurrent but showed better utility preservation (Section 2).
* sL85 asked about relation to RLKF → we clarified that RLKF does not target open-ended hallucination reduction.

**Miscellaneous points**

* sL85 asked about GRPO vs PPO or Reinforce++ → we clarified that the method is RL-algorithm-agnostic.
* q2wP requested evaluation beyond Qwen → we added results on Llama-3.1-8B (Tulu3) (Section A.3).

---

We hope this summary makes the contributions and revisions clear. We sincerely appreciate the reviewers’ careful feedback and thank you for considering our responses and the strengthened version of the paper.

Best,

Authors of paper #22292

---

### Meta-Review · Area_Chair_hDhj · 2025-12-09

**Summary:**

This paper applies reinforcement learning methods to mitigate hallucinations in large language models.

### Pros
* The binary reward scheme is straightforward and easy to implement.
* Experiments show the method effectively reduces hallucination rates.
* The experiments are thorough, and the work is relatively comprehensive.
### Cons
* Incremental novelty
* The method is conceptually very similar to Reinforcement learning from knowledge feedback
* Evaluation limitations.
* The unfairness of baseline comparisons

### AC's evaluation.

1. From reviews and rebuttals

The consensus is negative (4, 6, 2, 2). Reviewer q2wP (2) strongly rejects the paper, arguing that the mechanism is just making the model conservative and shorter, and the contribution is poor.
Reviewer eTuW (2) highlights methodological flaws in how "I don't know" is trained and evaluated.
Reviewer G4jz (4) rejects based on incremental novelty.
Reviewer sL85 (6) weak support, but flagged similarity to RLKF. The authors failed to provide a convincing explanation for why binary rewards are theoretically superior beyond just acting as a heavy regularizer for conservatism.

2. From AC's reading

The paper presents a useful engineering observation: binary rewards make RLHF more stable and conservative. However, this does not constitute a significant scientific contribution. The improvement in hallucination metrics appears to be largely a trade-off with content recall (conservatism), which the paper does not adequately analyze. Combined with the lack of novelty relative to VeriScore and RLKF, the paper is not ready for publication at ICLR. Luckily, most reviewers share the same points with AC, so the rejection is reasonable.

**Reviewer Concerns:**

Resolved Concerns:

1. Mechanism of Reward Hacking: The authors explain why dense rewards lead models to generate nonsensical text to cheat for points by analyzing reward distributions. Reviewers largely agree with this explanation.

Outstanding Concerns:

1. Lack of Novelty (Reviewer G4jz): Merely discretizing Soft Rewards into Binary values is deemed lacking in substantive innovation. The authors failed to demonstrate deep insights behind this modification, stopping at merely “experimental effectiveness.”

2. Conservatism vs. Capability (Reviewer q2wP): The reviewer contends that the improvement stems from the model becoming overly conservative (over-refusal) rather than genuine capability enhancement. The authors failed to provide detailed analysis on Recall to counter this argument.

3. Evaluation Robustness (Reviewers eTuW & sL85): Relying on string matching to identify “I don't know” responses is deemed inaccurate. Moreover, the training objective forces the model to ignore parameterized knowledge, potentially causing it to “know but say it doesn't know”—an incorrect alignment direction.

4. Fair Comparison (Reviewer eTuW): The lack of a rigorous comparison with VeriScore under identical KL constraints fails to demonstrate the inherent superiority of Binary Reward itself.

**Reviewer Scores:**

Reviewer q2wP:  deemed contribution extremely low, mechanism poorly explained (model becomes overly conservative), definite rejection.

Reviewer eTuW identified flaws in experimental design (training objective and evaluation method), unfair comparisons.

Reviewer G4jz:  Innovation is a major flaw; deemed a “fine-tuning” approach.

Reviewer sL85:  Method is simple and effective, but also points out similarities to RLKF; support is very weak.

All reviewers will maintain their scores.

---

### Decision · Program_Chairs · 2026-01-26

Reject